# Emergent Aspects of the Integration of Sensory and Motor Functions

**DOI:** 10.3390/brainsci15020162

**Published:** 2025-02-07

**Authors:** Tiziana M. Florio

**Affiliations:** Department of Life, Health and Environmental Sciences, University of L’Aquila, 67100 L’Aquila, Italy; tizianamarilena.florio@univaq.it

**Keywords:** sensorimotor integration, motor control, imagery, rehabilitation

## Abstract

This article delves into the intricate mechanisms underlying sensory integration in the executive control of movement, encompassing ideomotor activity, predictive capabilities, and motor control systems. It examines the interplay between motor and sensory functions, highlighting the role of the cortical and subcortical regions of the central nervous system in enhancing environmental interaction. The acquisition of motor skills, procedural memory, and the representation of actions in the brain are discussed emphasizing the significance of mental imagery and training in motor function. The development of this aspect of sensorimotor integration control can help to advance our understanding of the interactions between executive motor control, cortical mechanisms, and consciousness. Bridging theoretical insights with practical applications, it sets the stage for future innovations in clinical rehabilitation, assistive technology, and education. The ongoing exploration of these domains promises to uncover new pathways for enhancing human capability and well-being.

## 1. Introduction

A fundamental characteristic of human sensory experience relies on the “unified” percept, although sensory information comes from different sensory modalities. This integration derives from the ability to combine multiple sources of sensory information by different neural circuits. Multisensory responses of superior colliculus neurons, and multisensory integration, first observed in the superior colliculus of cats [1], have now been recognized as a common feature of numerous brain areas of different species, both developmentally and evolutionarily. Multisensory integration of cross-modal signals has been described in the prefrontal cortex, in which neurons are sensitive to both the synchrony and semantic context of vocalizations, but also in premotor, parietal, and temporal cortices. Multisensory transformation has raised new ways to understand computational principles like Bayesian inference, through which it has been possible to link multisensory integration (priors) at the level of neurons and behavior (regularities learning). The perception, thus, is determined by priors and sensory likelihoods; in turn, priors reflect the accumulated history of sensory likelihoods. Interestingly, the evolution of multisensory integration models and the evaluation of fake sensations, like the rubber hand illusion and the full body illusion (virtual reality), have led to questions about peripersonal space and bodily self-consciousness, opening up new ways to understand and approach psychopathologies like schizophrenia and autism spectrum disorders, among others [2,3].

In the following section, a brief description of recent advancements in sensory integration and sensorimotor integration will be presented.

## 2. Sensory Integration

Traditionally, the sensory system has been described as a system in which each of the five senses has its own specific receptors, dedicated pathways, and target regions. This concept, known as the “Law of Specific Nerve Energies”, posits that each sense operates independently, following its own “labeled line” [1]. However, subsequent research has revealed that different submodalities and cross-modal information can be spatially integrated and temporally summated, leading to the concept of multisensory integration [4]. This paradigm shift has advanced our understanding of how sensory processing generates more coherent activation patterns across large-scale networks, thus enhancing the efficiency of information processing. Tactile signals, for example, are transmitted from the skin to the spinal cord and brainstem via two pathways: the “direct” (paucisynaptic or monosynaptic) and “indirect” (polysynaptic) routes. Recent evidence shows that these pathways converge onto the same brainstem neurons, supporting the idea that tactile sensation is produced by a combination of inputs [5], as observed in other sensory systems [6]. A recent study has described a convergent input to the brainstem inferior colliculus deriving from skin mechanoreceptors and auditory input. The coincidence of dual sensory inputs is proposed to allow brainstem centers to mediate behavioral responses to environmental vibrations [7]. The degree of convergence onto individual brainstem relay neurons determines the type of information transmitted to different brain regions, challenging the notion that cortical neurons should be classified based on their function rather than their submodality composition [8]. The discovery that specific sensory and multisensory neurons possess overlapping receptive fields was a significant step in advancing the understanding of subcortical structures, such as the superior colliculus and thalamus, in sensory processing.

### 2.1. Superior Colliculus Functioning

The superior colliculus integrates multiple sensory modalities, including visual, auditory, and somatosensory information, and combines them with motor commands to produce spatially tuned behaviors [9,10]. Highly conserved across vertebrate evolution [11], the superior colliculus has played a crucial role in visual processing for gaze orientation. The superficial layers of the superior colliculus represent visual space, while neurons in the deep layers respond to auditory and somatosensory stimuli [12]. Distinct sensory inputs are mapped onto spatially aligned somatotopic maps. This organization ensures that signals originating from the same event, which are spatially and temporally concordant, converge at the same point on the motor map [13]. This alignment facilitates sensorimotor transformations that are essential for the superior colliculus’ role in detection, localization, and orientation behavior [1]. Based on the observation that the visual receptive fields in the superficial layer of the superior colliculus are aligned with the movement vector map of the intermediate layer of the superior colliculus, a number of visuo-motor transformations have been described [14,15]. The superior colliculus exhibits a complex intrinsic organization and extensive extrinsic connectivity with key structures such as the thalamus, basal ganglia, amygdala, and related cortical areas [13,16,17]. These connections contribute to its role in visual-guided behavior such as spatial orientation, motion detection, and biological movement recognition, but also attentional and decision-making superior functions [11,18,19]. As recently refined by [17], the mouse SC is organized into distinct parallel subnetworks throughout the brain, each associated with different functional specializations, such as orienting movements during exploratory vs. consummatory behavior, or defensive vs. aggressive behavior. Different subregions and neuronal populations functionally interact with other brain regions to drive complex behaviors. Multiple sensory information from diverse peripheral receptors is distributed across different collicular layers to be spatially and temporally aligned so that features or events occurring together in time and/or location can be readily integrated. This integration, when cortically integrated, leads to response enhancement (or depression) characterized by neural gain, meaning that paired stimuli elicit neural responses greater (or less) than what would be expected from the sum of the responses to individual stimuli [20], particularly at the beginning of the responses [21]. Superadditivity has been proposed as the gold standard for measuring multisensory integration [22]. Interestingly, similar additive, superadditive, or suppressive mechanisms can be elicited in human cortical areas, such as the superior temporal sulcus, mediating efficient integration between degraded or intact stimuli during object categorization [23].

### 2.2. Thalamus Functioning

All sensory information, except for olfaction, passes through thalamic relay centers before reaching the sensory cortex [24]. The thalamus mediates interactions between cortical and subcortical systems. Reciprocal connections exist between the cortex and the thalamus that enable the thalamus to function as an integrative hub [25]. Using different methodological approaches, several studies [25,26,27] converge on the idea that the human thalamus operates as an integrative hub [25]. Its specific connections with sensory-specific cortical areas are implemented by extensive connections with most functional networks [26], playing a crucial role in influencing (or organizing) the large scale of brain network processing [27]. Therefore, rather than being a relay station, the human thalamus is critically involved in shaping key functions such as arousal, cognition, and conscious awareness [27]. First-order thalamic nuclei receive their primary inputs from subcortical sources. The higher-order nuclei are primarily driven by cortical inputs [28]. The thalamic reticular nucleus serves as the principal source of intrathalamic inhibitory input. However, the characterization of its organization into distinct tiers, shaped by excitatory input, suggests that thalamic reticular neurons play a role beyond simple feedback inhibition to relay cells [29]. The thalamic reticular nucleus functions as a “searchlight” for sensory input, integrating cortical and subcortical signals to selectively regulate thalamic relay cells, ensuring that the appropriate information reaches the cortex [29]. All sensory nuclei within the thalamus exhibit modality-specific topographical organization. External inputs to the thalamus modulate thalamic firing patterns via the thalamic reticular nucleus, inducing distinct oscillatory rhythms that regulate brainwave activity and states of arousal [30]. Sensory information relayed from the thalamus to the cortex maintains the modality-specific topographic organization, ensuring accurate sensory mapping. In contrast, higher-order thalamic neurons, along with first-order neurons, project extensively to multiple cortical areas [31]. This projection pattern forms an indirect trans-thalamic communication pathway between cortical regions, facilitating cross-modal sensory integration [32]. Consequently, individual sensory cortical areas can receive input from a diverse array of thalamic nuclei. Corticothalamic inputs originate from various cortical regions and project to subcortical structures, such as the superior colliculus and the basal ganglia [13,33]. These reciprocal connections between the cortex and thalamus underpin the spatial and temporal modulation of incoming sensory information. Spatial modulation is governed by the feedback and feedforward interactions between corticothalamic and thalamocortical pathways. The topographic organization and spatial specificity of corticothalamic fibers enable mechanisms like surround suppression, where the center-surround retinotopic excitatory receptive fields are flanked by larger regions that suppress responses to visual stimuli [34]. Cortical feedback can amplify and refine these responses, a process observed in both the auditory and somatosensory systems. Such modulation also occurs in the reverse direction. Through dynamic, bidirectional interactions between feedforward thalamocortical inputs and feedback corticothalamic inputs, sensory tuning is achieved [35]. Temporal modulation is mediated by the Thalamic Reticular Nucleus [30]. When corticothalamic neurons fire at low frequencies, their activity is transiently enhanced before being suppressed by Thalamic Reticular Nucleus -mediated thalamocortical inhibition. In contrast, the high-frequency firing of cortical neurons leads to short-term facilitation, while Thalamic Reticular Nucleus-mediated thalamocortical synapses undergo short-term depression [36]. This mechanism allows Thalamic Reticular Nucleus -induced depression to be overcome, resulting in a sustained enhancement in thalamic activity, constrained within specific temporal windows [37]. The temporal patterning of thalamic inputs significantly influences the cortical discrimination and detection of sensory stimuli [36].

### 2.3. Synchronization Drives Integration

The spatial and temporal principles of multisensory integration suggest that the closer two unisensory stimuli are (in space or time), the more likely they are to be integrated into a unified percept [3]. Beyond processing, stimuli coming from various cortical networks must be integrated with each other to form a coherent perception. The integration of information across sensory modalities occurs through the transient synchronization of neural oscillations [38]. Oscillatory neural activity reflects the synchronized fluctuations in the summed postsynaptic activity of large neuronal populations. These oscillations can be broken down into distinct frequency components, each defined by specific amplitudes and phases [39]. Various types of oscillatory responses, each linked to different forms of neural synchronization, can be identified. Research has shown that bottom-up processing primarily involves local networks operating in the gamma frequency band [>30 Hz], while top-down control, involving long-range integrative processing, engages lower frequency bands [<30 Hz] [40]. Multisensory processing depends on the coordinated activity of different cortical regions establishing a causal relationship between brain oscillations and behavior [41]. Key mechanisms include local neural oscillations and functional connectivity between distant cortical areas [42]. Emerging evidence indicates that neural oscillations in distinct frequency bands correspond to specific mechanisms in multisensory processing [38]. It is proposed that oscillations in different frequency bands play roles in feedforward–feedback processing, attention modulation, and predictive coding during multisensory integration. These mechanisms may also operate concurrently [43]. Through the process of multisensory integration, not only are the external environment and the observer’s body unified, but the individual who perceives the environment and the agent that interacts with the world are also unified. In other words, multisensory integration plays a crucial role in shaping human–environment interactions from both the “environment” and “individual” perspectives. By expanding the discussion, the process of multisensory integration is rooted in the structure of bodily self-consciousness [3], and can be applied to learning and education and the expression of cultural heritage when considered as a significant aspect of society [44]. The combination of signals from different sensory modalities can enhance perception and facilitate behavioral responses. In essence, the synthesis of complementary information coming from multiple sensory modalities improves perceptual performance, leading to more accurate and faster responses. Crossmodal interactions are governed by neural oscillations in different frequency bands that can occur at both early and late stages of processing, and involve bottom-up and top-down mechanisms. Recent studies [45] have examined the crossmodal facilitation of response speed, as manifested in the speed of visual responses to concurrent task-irrelevant auditory information. These data reported that the crossmodal facilitation of response speed was associated with reduced early beta power in multisensory association and secondary visual areas. This beta power modulation has been supposed to reflect an early stage of behaviorally relevant crossmodal feedback influence from higher multisensory areas and secondary visual areas, improving visual processing through attentional gating [45]. In both the multisensory integration and the crossmodal learning of complementary information, the contribution of each sense to forming a multisensory representation of an object depends on several factors. These include the range within which a sensory modality works best, how reliable or unique the sensory information is, and expectations based on prior knowledge. Additionally, beyond the direct influence of sensory input on perception, learning features across different senses can impact how information is stored in memory. This, in turn, can affect how objects are classified and recognized [46].

## 3. Sensorimotor Integration

The integration of sensory and motor functions underlies the brain’s ability to plan and execute actions based on environmental cues. This process involves the continuous updating of motor plans based on sensory feedback, ensuring that movements are accurate and goal-directed. The ability to adapt motor actions in response to changing sensory information is essential for effective interaction with the environment. The execution of goal-oriented behaviors requires a spatially coherent alignment between sensory and motor maps. Recently, a direct connection between collicular motor units and kinetic visual features has been described that enables the goal-directed behavior organizer to rapidly intercept moving and static targets [47]. Multisensory processing is critical for coordinating complex motor tasks. The brain integrates information from multiple sensory sources, such as vision, hearing, and touch, to create a comprehensive understanding of the environment [48]. This multisensory integration allows for more accurate and efficient motor control, enabling individuals to perform tasks that require the coordination of different sensory modalities. Sensory inputs play a vital role in guiding motor actions, with parieto-frontal circuits mediating visuomotor transformations [49]. The brain integrates visual, auditory, and somatosensory information to guide movements, ensuring that sensory information is effectively translated into motor commands. Specific cortical areas are involved in visuomotor integration. The ventral intraparietal area (VIP) and the anterior intraparietal area (AIP) are reciprocally connected with the F4 and F5 premotor areas of the monkey’s frontal cortex. The VIP-F4 circuit is responsible for movements of reaching, approaching, and withdrawing with the arm, trunk, and face/mouth [50]. Similarly, the AIP-F5 circuit guides visually directed grasping movements, integrating visual information about objects with motor commands to the hand of monkeys [51,52,53]. The largest part of the parietal cortex plays a crucial role in integrating sensory inputs with motor plans. It receives and processes information from various sensory modalities, creating a cohesive representation of the external environment. This information is then transmitted to motor areas, where it is used to guide actions. Due to interactions within the corticothalamic systems, the brain exhibits rhythmic neural activity [54]. Various oscillation patterns are associated with different brain functions, ranging from sensory and motor to cognitive tasks. Although there is not complete agreement [55], the phase of oscillations influences brain function, and the synchronization of oscillations within specific frequency bands—known as the communication-through-coherence model—can alter perception, attention, and working memory [41]. For instance, alpha oscillations in the parietal area are linked to controlling visual perception across different spaces [56]. In contrast, beta and theta oscillations in the parietal region are involved in perceiving local versus global image features, with theta and beta oscillations, respectively, associated with the encoding of this information [57]. These observations support the idea that brain oscillations facilitate a complex pattern of information processing and communication between different brain areas [41]. Beta oscillations with frequencies ranging from 13 to 30 Hz are thought to be involved in somatosensory processing and motor control, and have been predominantly observed in sensorimotor cortices and the basal ganglia. The origin of beta synchrony has been extensively discussed and associated with oscillations of the whole basal ganglia–cortical loop [58,59]. Changes in beta oscillation power have been functionally linked to phases of movement [60]. Typically, beta oscillations are prominent during stable postures and diminish during active states, such as movement planning and execution [58]. Beta oscillation power is spontaneously and bilaterally reduced in the premovement period and increased close to the premovement period [60]. The gamma oscillation, in contrast to the beta one, undergoes a decrease after practice [61]. Similarly, the motor cortex shows a decrease in beta power during ipsilateral execution movements [62]. More specifically, support for the emergence of synchronous activity in the beta-band associated with pallido-striatal connections during static conditions, and in the gamma-band associated with pallido-subthalamic feedback loops during movement, have been provided in a silica-based model [59]. The level of beta synchrony has been found linked to dopamine levels in the corticobasal ganglia loop, which is probably related to salient internal and external cues [63,64]. Parkinson’s disease, a hypokinetic movement disorder resulting from the loss of nigral dopaminergic input to the striatum, is associated with the disruption of the beta-band oscillatory firing of the globus pallidum, subthalamic nucleus, and pars reticulata of the substantia nigra [64,65,66,67]. These dynamics in the basal ganglia can contribute to the desynchronization of the beta frequency band in the motor cortex through the thalamocortical pathway [62]. A notable reduction in beta oscillation amplitude also occurs in sensorimotor areas during motor imagery or when observing actions [60]. In conditions where motor function is functionally decreased, as in aging, or pathologically impaired, such as in Parkinson’s disease, there is a corresponding disruption in beta oscillations, indicating an altered motor release. These oscillations have different roles, including reducing inhibition to initiate motor plans, maintaining current motor states against internal and external noise disruptions, and facilitating motor learning [60]. The hierarchical organization of motor control systems ensures that simple movements engage minimal cortical areas (partial autonomy of lower-level motor control [68]), while complex movements activate broader networks, including the supplementary motor area, premotor cortex, parietal cortex, basal ganglia, and cerebellum. This hierarchical control allows for the smooth execution of movements and the ability to adapt to new and changing environments.

In the following section, a brief description of the principal subcortical and cortical areas involved in motor control and sensorimotor integration is presented.

## 4. Basal Ganglia and Sensorimotor Integration

Most voluntary actions necessitate choosing a single action from a broad array of potential motor programs. This intricate process of action selection involves merging various signals, such as sensory and contextual data from the surroundings, previous experiences, and self-generated goals. All these information often conflict, requiring a system to choose the most adaptive and suitable action for varying circumstances. Such a system would need to preserve and individually represent both goals and signals, allowing for their independent modification and manipulation, and determine and execute one action from among the alternatives through downstream motor structures. The basal ganglia are crucial in modulating these functions through both ascending and descending pathways. Receiving input from virtually all cortical areas, the striatum is located at the interface for mediating cortico–subcortical interactions [69,70]. The striatum converts a variety of signals from the cortex—including sensory, cognitive, and motor signals—into action selection cues that either inhibit or activate relevant motor programs within specific motor modules [71,72,73,74,75,76,77]. Receiving dopaminergic inputs from the pars compacta of the substantia nigra, the striatum processes the environmental stimuli marked as significant ones during motor learning [69,72,74]. The striatal internal structure, including the “direct” and “indirect” pathways identified by Alexander and Crutcher [78], along with the “striosomes” and “matrix” compartmentalization [79], illustrates the intricate framework of basal ganglia processing. This complexity is further detailed by the “funneling” of the extensive cortical inputs into sensorimotor, limbic, and cognitive parallel loops [80]. These loops have been thought to originate from and return to the same input sites, showcasing the extensive potential for processing within the basal ganglia. On the other hand, it has been recently described that the dorsolateral striatum receives excitatory inputs from both sensory and motor cortical regions [81]. Both motor and sensory information are modulated by direct and indirect corticobasal ganglia–thalamo–cortical pathway, and are altered by dopamine depletion [81,82]. The direct pathway acts by disinhibiting the thalamus, ultimately facilitating the motor behavior being executed. The indirect pathway acts by promoting the tonic suppression of thalamic activity, thereby inducing the cessation of motor activity. According to this model, a selective loss of indirect projections would result in an increase in spontaneous involuntary movements, as seen in hemiballismus. Conversely, an increase in the neuronal activity involved in the indirect pathway, along with the loss of direct connections, would lead to a reduction in thalamocortical facilitation, resulting in the bradykinetic and akinetic condition characteristic of parkinsonism. A third “hyperdirect” pathway connects the subthalamic neurons with the cortical inputs, bypassing the striatum, and excites pallidal output neurons, increasing their inhibitory output (Figure 1). The functional property of the hyperdirect pathway to inhibit a cortical motor program would enable the direct pathway of the basal ganglia to adjust the motor program according to the situation [83,84], and the subthalamic nucleus to dynamically control the effective decision limit over time [85]. The dopaminergic neurons of primate’s substantia nigra exhibit increased firing activity in response to “significant” sensory stimuli. This neuronal activity is linked to the nature of the reward that the animal expects to receive after executing a given command, and is not directly related to the movement itself. The D1–D2 dopaminergic receptor subtypes are specifically inserted on striatal projecting neurons of the direct and indirect pathway and are, respectively, excited or inhibited when a significant stimulus evokes a shot of dopamine from the pars compacta of the substantia nigra [86]. These responses gradually diminish as the animal learns and automates the task. Thus, the dopaminergic system of the basal ganglia, by transmitting signals indicative of the incentive value of a given environmental stimulus, influences the acquisition of the conditioned response at the striatal level and, ultimately, modulates the behavior at the onset of learning and over time with the maintenance of the “success” that behavior has produced [87]. Striatal interneurons are actively involved in these processes. They gradually acquire a conditioned excitatory response as conditioning progresses, which is retained in long-term memory and used when the conditioned behavior is fully acquired and automated. Cholinergic interneurons in the striatum, also named tonically active neurons, detect coincidences between sensory inputs and behavioral context, adjusting the sensitivity of other striatal neurons to cortical inputs (Figure 1) [88,89,90]. Therefore, the sensory responses processed into the striatum can be modulated by motor activity thus accounting for the ability of the basal ganglia to control movement in relation to the contextual variability [81]. The sensorimotor cortical regions and the dorsolateral region of the human putamen are suggested to be the long-term storage structures where well-learned movement sequences are preserved. During motor learning, a concordant shifting of activations from anterior to posterior regions of basal ganglia that mirror changes in the motor areas of the frontal cortex has been described. Initially, new learning activates the dorsolateral prefrontal cortex and striatum; selected movements engage the premotor cortex and mid-putamen; and automatic movements activate the sensorimotor cortex and posterior putamen. Focusing on well-practiced actions, activation returns to the dorsolateral prefrontal cortex and striatum. Notably, the cerebellum remains inactive during new decisions, focused actions, or when selecting movements [91].

The consistent projection relaying the basal ganglia with prefrontal areas implied in cognitive processing and the mirror system suggests that the basal ganglia contribute to higher levels of cognitive functions, such as planning, syllogistic reasoning, and mathematical problem solving. To this regard, using a gradient functional connectivity approach O’Rawe [92], have recently described a double orientation of human caudate functional relations with cortical networks, confirming previous non-human data. A medial organization of these functional relations has been associated with internal orienting behavior, whereas the lateral extent of the caudate is more related to externally oriented behavior [93]. Notably, recent evidence includes clustered representations of five separate higher-order association networks within the caudate, positioned side-by-side and linked to higher-order networks underlining language and social functions. These clusters are spatially separate and asymmetrically distributed, paralleling the cortical lateralization [94].

## 5. Cerebellum and Sensorimotor Integration

The other important subcortical hub for sensorimotor integration is represented by the cerebellum that might function by expanding the range of sensorimotor associations, in accordance with the dynamics shaping the behavioral condition, while adjusting the input–output coupling rules to fit the broader context [95]. Classically, the cerebellum has been primarily involved in motor learning, including the error detection and correction that are essential for refining motor skills over time. When a new motor act is performed, the cerebellum adjusts the timing and intensity of muscle activations to improve performance prevalently, based on response feedback. With repetition and practice, these adjustments become more precise, resulting in smoother and more coordinated movements. Traditionally, two models have been used concerning cerebellar motor functions: the theory of “cerebellar control mechanism” [96], and the “cerebellar motor learning” mechanism [97,98]. Briefly, the motor learning model would function by modulating the responsiveness of Purkinje cells, thereby serving as a teaching or error-correction signal [99]. According to Llinás’ model [100], the cerebellum is believed to function as a control mechanism, utilizing the rhythmic activity of the inferior olive to synchronize Purkinje cell populations, thereby fine-tuning coordination [98]. The most accepted role of cerebellum has classically regarded the “error-encoding” function played by the Purkinje cells system under the climbing fibers as driving information about sensory information incongruency as “complex spiking”. This way, the cerebellum functions as a comparator of inferior olive signals with respect to the cortical upcoming commands, enabling the corrections of motor plans whenever incongruences between them are detected [101]. This mechanism implies that the cerebellum, developing new input–output associations, is responsible for motor learning. Thus, motor learning depends on cerebellar ability to map outgoing motor commands to the predicted sensory feedback resulting from those movements. The ability to learn through sensorimotor adaptation may be linked to the resulting impairment in updating motor commands in patients affected by cerebellar disorders [102]. On the other hand, findings have shown that climbing fibers convey a variety of non-error signals, too. It has been observed that the “simple spike” activity of Purkinje cells predicts performance errors and conveys feedback signals related to movement kinematics. This perspective has strengthened the notion of a forward internal model represented by a simple spike, which is updated based on changes in the behavioral state through input from climbing fibers [103]. Beyond the online motor control, cerebellar processing has been reported for the persistent representation of information in motor areas of the frontal cortex. During motor planning, preparatory activity has been described in both the cerebellum and the motor frontal cortex. After cerebellar processing, efferent projections reach the frontal cortex again through the thalamus, closing a cortico–cerebellar loop that is able to work in the time-scale of seconds [104]. Indeed, Boven and Cerminara [105] have found that the ability of the cerebellum to support millisecond timescales might be intrinsic to cerebellar circuitry, and the ability to support supra-second timescales might result from cerebellar interactions with other brain regions, such as the prefrontal cortex. This way, the cerebellum makes predictions (Figure 2) and provides feedback over multiple time horizons, spanning from milliseconds when engaged in motor control to seconds or more when involved in decision-making or long-term motor planning [106].

Hence, the neural mechanisms underlying sensory integration and motor control involve the coordinated activity of various cortical and subcortical regions. All these areas are highly interconnected, with extensive communication pathways that facilitate the integration of sensory and motor functions from planning to execution. The parietal cortex, the premotor cortex, and the cerebellum work together to process sensory inputs and generate appropriate motor responses. The prefrontal cortex, the supplementary motor area, and the basal ganglia work together to process inputs and respond to demanding environments. The functional dissociation between these two internally vs. externally guided loops has been broadly studied in the control of movements [108].

## 6. Cortical Areas and Sensorimotor Integration

Motor planning and control rely heavily on the brain’s ability to predict the outcomes of movements and adjust subsequent actions accordingly. This predictive framework involves the representation of the intended movement and its consequences, thus forming the basis for abstract behaviors and logical movement sequences [107]. Motor schemas, which are sequences of muscle activations necessary to perform an action, are learned through experience and stored in procedural memory. The premotor cortex and supplementary motor area play critical roles in representing and executing movements (Figure 3) [109].

During the initial stages of motor learning, the premotor cortex, with its extensive connections to the parietal cortex, integrates external information to develop new motor programs [111]. As the expertise becomes more refined, motor control shifts to the supplementary motor area, which is involved in executing learned and internally driven movements [112,113]. The hierarchical motor control system engages various cortical and subcortical regions, ensuring smooth and adaptive motor performance. The primary motor cortex is primarily responsible for the detailed execution of movements, while higher-order areas like the premotor cortex and supplementary motor area plan and coordinate complex actions. The cerebellum and basal ganglia, as mentioned, are crucial for refining motor skills and ensuring precise execution [114,115,116]. The cerebellum is involved in timing and coordinating movements, detecting and correcting errors, and adapting motor commands based on sensory feedback [106]. The basal ganglia facilitate the selection and initiation of appropriate motor programs, playing a critical role in procedural memory and motor learning [69,72]. The medial prefrontal cortex and the striatum are involved in matching results of behavior with expectation based on a predictive ability to represent the consequences of an action [117]. To this regard, it has been recently demonstrated that during learning of a rewarded task, the default mode network (DMN) and dorsal attention network (DAN) underwent shifting of their connectivity. At the beginning of the learning process, an increased functional coupling between several sensorimotor areas of the DAN (premotor cortex) with salience/the ventral attention network (anterior insula/inferior frontal gyrus and anterior cingulate cortex) was recorded. On the contrary, during late learning, those DAN sensorimotor areas switch their connectivity to DMN areas (medial PF cortex), detecting a transition in the functional coupling of the sensorimotor areas to the transmodal cortex when adapting behavior. This shift might reflect a transition from more exploration to exploitation activity when decisions can be made based on established sensorimotor contingencies [118].

Predictive representations in the brain enable the anticipation of movement outcomes, which is essential for adjusting actions in real-time. The cerebellar role in error detection and correction involves monitoring the accuracy of movements and making necessary adjustments to improve performance, contributing to new motor learning [119]. This process is vital for learning new motor skills and refining existing ones, particularly based on the ability of the cerebellum to respond to reward-related error signals [120]. The ability to adapt motor plans based on sensory feedback is a hallmark of effective motor control. The integration of sensory inputs with motor commands allows the brain to continuously refine and optimize movements, ensuring that actions are goal-directed and efficient. This adaptability is crucial for responding to changing environmental conditions and achieving complex motor tasks. The parallel and synchronized interaction between cortical and subcortical structures allows the mental representation of action and its consequences, the comprehension of other’s motor acts, and temporal abstraction. A huge of learning practice, clinical therapeutics, and technological applications have been started from these perspectives. Relevant in this last area is the concept that synthetic systems, like AI-based devices, need to “learn” from motor behavior, too. The relationship between imagination and reality is evident in the immediate neural adaptations that accompany physical exercise, but also in motivational and affective outcomes [121].

## 7. Mirror System and Sensorimotor Integration

Because of what we have previously described, it appears difficult to clearly define a sharp separation between motor and sensory functions. Such a division has also vanished in the cerebral areas that are ever-engaged in motor control, such as the primary motor cortex. During the execution of a manual interception task in varying locations, the persistent activity of most monkeys’ neurons in the primary motor neurons showed that the sensory modulation of motor output probably related to predictive sensorimotor control [122]. The seminal study of Rizzolatti and coworkers [52,123] provided new insights about the complex organization of cortical and subcortical areas, traditionally implied in “pyramidal” and “extrapyramidal” loops, and their functional roles in sensorimotor integration. The posterior [F1–F5] parieto-dependent motor area relates to primary motor cortex and send direct projections to the spinal cord. The rich sensory information originating from the parietal lobe enables the parieto-dependent motor area to process sensory–motor transformation. Contrarily, the anterior [F6–F7] prefronto-dependent areas do not send fibers to the primary motor cortex and show diffused connections with other motor areas receiving higher-order cognitive information, relating this circuitry with motor planning and motivation. Particularly, the sensorimotor transformation by F5 neurons enable this area to categorize for specific actions, by which it is possible to plan movement starting by coding the action at different degrees of abstraction (prototypes of actions) that facilitates the association between sensory properties of the objects (affordances) and the appropriate motor plan to interact with these objects. These connections and functional evidence permitted the formulation of the principle by which there is direct coupling between an action’s observation and its execution. The F5 neurons exhibiting such a property have been collected as an integral part of the mirror system whose functional roles are implicated in action recognition and imitation [123]. As in monkeys, several fMRI, TMS, EEG, and electrophysiological studies have reported that a fronto-parietal mirror system also operates in humans [124,125]. The cortical areas including the inferior parietal lobule, the ventral premotor cortex, and the inferior frontal gyrus are involved in the goal and intention of the observed motor act. Based on the activation of inferior frontal gyrus, an intransitive movement mirror system has been described as the basis of the imitation capacity [126]. The left-sided human speech motor centers have been described as able to elicit motor representation of the heard phonemes in the corresponding motor representation of the same sound that is involved in the comprehension of word meaning [126,127,128]. Still, the representation of certain types of emotion has been found to activate the insula and cingulate sites able to evoke emotion and visceromotor expression of the same emotions [53,126,127]. These findings have raised the hypothesis that others’ emotions are recognized through the activation of the neural centers mediating the feeling of that emotions in the observer [49]. This way, specific body motor areas, hand and mouth principally, which representations are somatotopically replicated in different cortical areas (frontal, posterior parietal regions) are mirrored. The specific activity of some neurons placed in these cortical areas relates to the general goals of motor actions. The consequent categorizations of different motor actions [128] enable the nervous system to code different motor acts, to generate a repertoire of actions and to activate their motor representation whenever observing the movements of others. These mechanisms are at the basis of the ability to understand what the others are doing, mirroring the agent’s motion into the observer’s internal representation of that motion, that is to understand what goal the agent is reaching through that action, or to understand the intention of others [49,123,126]. Similarly, the aptitude to imitate others motor actions pertain to these mechanisms. Imitation learning, which consists of the ability to copy a behavior that is not previously present in the motor repertoire of the observer, is operated by dissecting the component of agent’s motion into string of component motor acts. The observer can “recognize” and code the sequence of each component in its own repertoire so that the new imitated actions become similar to the agent’s motion (ideomotor theory of action). The integration between internal imagery ability and external cues information enable to target different “goals”, attributing them with specific functional meanings [129].

Sensory and motor processing share the same substrates [130]. The integration of sensory and motor functions enables the representation of action, its outcome, and the updates resulting from the execution of an effective motor action [131]. As a consequence, actions and subsequent ones can be concatenated into routines. Concatenation relies on the anticipation mechanism, which is based on the ability to learn the causes of actions. Learning the consequences of actions allows for the control of the environment in a goal-directed manner [132].

The strength of a routinely tested concatenation of motor acts forms the basis of skill acquisition. Skills can, thus, be considered as a concatenation of motor routines whose outcomes have been “tested” against their representations, making their outcomes more predictable or their unpredictability manageable (categorization). These routines manifest in the form of automatic or habitual behavior or goal-directed behavior depending on the predictability of their outcomes and environmental characteristics [133]. The formation of outcome predictions is mediated by the intervention of corticocerebellar and/or corticobasal ganglia networks [134], which differ in their connectivity with cognitive networks. The medio-lateral functional organization of the basal ganglia [BG], depending on their differential connections with the default mode network [DMN], limbic, and attention networks, provide the anatomical substrate for the integration of prediction error and environmental information [92,93]. Moreover, evidence has been reported about the existence of direct communication between the basal ganglia and the cerebellum via mono- and disynaptic pathways [135]. The demonstrated connectivity between the subthalamic nucleus and cerebellar cortex, alongside reciprocal pathways from the dentate nucleus to the striatum, supports the hypothesis that these subcortical structures form an integrated functional network for adaptive learning [135]. This convergence suggests a dynamic framework where motor learning processes evolve over time, transitioning from initial cerebellar involvement in error correction and sensorimotor prediction to basal ganglia participation in habitual and reward-driven behaviors. This intricated picture has been recently highlighted by the relationship between the cerebellum and the mirror neuron system [MNS. The cerebellum contributes to modulating cortical inhibitory interneurons with mirror properties [136], thus emphasizing the cerebellar role in fine-tuning motor responses by learning models and predictive coding mechanisms. Such interactions, facilitating the integration of predictive modeling and reinforcement learning, enables the brain to refine motor commands through experience, optimizing movement execution based on past performance and expected outcomes. Moreover, the involvement of these structures in cognitive and affective domains suggests that the principles governing motor learning might extend to higher-order cognitive functions, offering new perspectives on how the brain integrates sensorimotor feedback to facilitate adaptive behavior across diverse learning paradigms. Disruption of these frameworks are related to impaired consolidation and refinement of internal model intervening during REM sleep [136].

In summary, the modality by which motor sequences are linked into new motor arrangements has been attributed to a distributed network composed by the presupplementary motor area, the supplementary motor area, the dorsal premotor cortex, the primary motor cortex, the primary somatosensory cortex, the superior parietal lobule, the thalamus, the basal ganglia, and the cerebellum. Motor schemas, learned through trial and error, become automated and are stored in procedural memory, enabling efficient and coordinated movements. During the early stages of motor skill learning, the dorsolateral part of the prefrontal cortex exhibits higher activity. As learning progresses through repetition, this activity is gradually replaced by the ventrolateral area of the prefrontal cortex. Skill acquisition through repetition is characterized by decreasing activity in the VL area and the increasing involvement of the orbitofrontal prefrontal cortex [137]. In the skilled sequential movements (those acquired after intensive and repetitive practice), the presupplementary motor area is involved in cognitive aspects of the early learning phase of movement sequences, and the supplementary motor area is involved in the performance of memorized movement sequences; meanwhile, the dorsal premotor area, the ventral premotor area, and the primary moor area are densely interconnected with each other to form a network for the control of hand movements. This organization seems to have the possibility to give the mirror system the cognitive link for internally guided movements and the temporal organization of the motor sequencing [138]. The expertise resulting from prolonged practice is associated with cognitive adaptation and the development of mental representation that captures the specific relationship between action and contextual demands, which are crucial for the organization of action control [139]. The resulting predictive capabilities about the outcomes of those actions play a crucial role in motor control, involving subsequent adjustments accounting for time estimation capability [140]. This prediction-based framework allows for abstract behaviors and logical movement sequencing, bridging the gap between sensory inputs and motor outputs.

## 8. Inferences from Sensorimotor Integration

The above-described multisensory and sensorimotor mechanisms of integration suggests that the brain is not merely a passive collector of external stimuli, but rather an active predictive machine that continuously generates expectations about incoming sensory data and updates these predictions based on new information. Perception and action machineries are deeply intertwined, forming an adaptive system that continuously updates internal models based on real-time interactions with the external world [141]. Sensory inputs dynamically influence motor responses, which in turn shape the acquisition and processing of subsequent sensory information forming a continuous feedback loop. This implies that cognitive processes cannot be understood in isolation from their interactions with the surrounding environment, emphasizing the concept of “embedded cognition”. Embedded cognition describes the idea that cognitive processes emerge from the dynamic interaction between the brain, body, and environment [3]. Notably, advances in multisensory integration models, including the evaluation of phenomena such as the rubber hand illusion and the full-body illusion in virtual reality, have provided insights into the mechanisms underlying peripersonal space and bodily self-consciousness. These insights have opened new avenues for understanding and approaching neuropsychiatric disorders such as schizophrenia and autism spectrum disorders, where disruptions in predictive processing may contribute to pathological symptoms [2,3,142].

The sensorimotor interplay is crucial for adaptive behavior, as the brain does not passively receive sensory data, but actively engages in predicting and refining sensory experiences. The embedded nature of cognition is evident in the interaction between perception and action, where motor outputs modulate the sensory inputs to optimize processing and response [143]. In such a context, attentional processes should be addressed. The embedded process model, as proposed by Cowan [144], suggests that attention operates within long-term memory representations, actively selecting and maintaining information based on the relevance and expectations set by the central executive system. According to this framework, sensory information can either enter the focus of attention through selective mechanisms or remain in the background as latent memory traces [145]. Specifically, attention enhances the precision of the sensory inputs by adjusting the weighting of prediction errors—discrepancies between expected and actual sensory signals. Focusing resources on the most relevant sensory data while filtering out less pertinent information, this modulation allows the brain to act as a predictive machine rather than a passive receiver of stimuli. These mechanisms are governed by hierarchical neural circuits. Structures such as the superior colliculus and pulvinar nucleus play a pivotal role in directing attention to spatially relevant stimuli, while prefrontal and parietal areas contribute to the modulation of sensorimotor loops [145]. Top-down influences from the prefrontal cortex guide attention toward relevant stimuli, facilitating efficient sensorimotor integration. In such machinery, precision signals, which are thought to be associated with neurotransmitter systems like dopamine, help regulate the balance between sensory-driven bottom-up signals and top-down predictions, ensuring that perception remains accurate and adaptive [141]. Attentional processes facilitating the sampling and updating of sensory–motor information have been associated with neural oscillations, particularly theta–gamma coupling. This dynamic interplay allows for the selection, maintenance, and integration of sensory inputs, enabling the brain to prioritize information that aligns with existing internal models while efficiently updating these models in response to unexpected inputs [143]. Then, attentive mechanisms, through rhythmic brain activity, ensures that relevant stimuli are processed in a temporally efficient manner, ultimately guiding perception and behavior in a goal-directed fashion.

Through the high-dimensional dynamics of the brain, the mu rhythm—oscillatory activity within the 8–13 Hz frequency range originating from the sensorimotor cortex—has been implicated in multiple brain regions and associated with emotional and attentional regulation [146,147,148]. Recorded both during effective execution and motor observation, the mu rhythm plays a critical role in sensorimotor integration by providing a neurophysiological link between action execution and observation [149,150], and has been associated with both motor imagery and motor execution [151]. As an indicator of sensorimotor resonance [152], mu rhythm desynchronization has been proposed as a potential biomarker of the mirror neuron system in both healthy individuals and those with neurological impairments [146,152,153,154]. While mu suppression primarily originates from the sensorimotor cortex, it also involves interconnected regions such as the premotor cortex and the inferior parietal lobule, which are essential for action understanding, motor planning and perception level [148]. Mu desynchronization is hypothesized to reflect the predictive mechanisms underlying sensorimotor integration within the mirror neurons system, facilitating the updating and reinforcement of internal motor representations based on observed actions. Accordingly, mu rhythm suppression during action observation has been identified as a potential early predictor of motor recovery, particularly in individuals with subcortical stroke. The extent of mu rhythm attenuation during action observation has been found to correlate with the degree of functional motor recovery, suggesting that the greater engagement of the mirror neuron system may enhance neuroplasticity and improve rehabilitation outcomes [155]. Consequently, mu rhythm-based interventions, such as action observation therapy and neurofeedback training, may be leveraged to optimize post-stroke motor rehabilitation. By specifically targeting the mirror neurons system and promoting sensorimotor integration, these approaches could facilitate the reorganization of cortical networks and improve functional recovery [156].

All key mechanisms of multisensory, sensorimotor integration and mirror systems are not only “embedded” within both the body and the environment, but are also deeply influenced by them. Their primary function is to mediate between internal and the environmental events, by the body, translating external and bodily inputs into different order of relations or connections (in terms of their properties and interdependencies). As such, they rely on shared processes to adapt to a dynamic environment, wherein expectations are continuously updated based on past experiences. In other words, they are all grounded in the incorporation of prior information, enabling the systems to learn new statistical regularities through behavior and generate predictions accordingly.

All these transformations in inferential knowledge have introduced novel perspectives for understanding computational principles. The Bayesian inference has been well suited in interpreting the functional logic of neuronal computations [2,3]. Thus, allowing for the linkage of multisensory and sensorimotor integration at both the neuronal and behavioral levels through the accumulation of sensory likelihoods over time [157]. Perception is thus determined by the interplay of priors, which reflects the history of sensory experiences and current sensory likelihoods, forming the basis for adaptive sensorimotor responses [157]. Bayesian methods integrate prior knowledge with observed data, allowing for the cumulative evaluation of evidence. Bayesian analysis generates probability distributions that provide metrics such as posterior means or credible intervals. In addition, Bayesian hypothesis testing evaluates posterior model probabilities, which helps to directly assess the strength of theories [142]. The idea that the brain can be viewed as a Bayesian inference machine that continuously generates predictions about the external world and refines them based on sensory feedback, striving to minimize prediction errors at multiple hierarchical levels, has been formalized as predictive coding framework [158,159,160,161]. Very interestingly, the predictive coding framework offers a unified explanatory model for various neurological disorders, highlighting how impairments in perceptual and motor inferences stem from the dysfunctional integration of sensory and motor information. For instance, in Charles Bonnet syndrome, the absence of adequate sensory input would prevent the correction of perceptual hypotheses, leading to complex visual hallucinations. Similarly, in anosognosia for hemiplegia, the inability to compare motor predictions with actual sensory data would result in a lack of awareness of paralysis. Furthermore, a similar disruption of the likelihood mapping between two regions has been hypothesized in conduction aphasia, which is anatomically consistent with a human mirror neuron system [162]. The examination of the role of mirror system in inferring intensions from action observation from the predictive coding perspective suggests a possible independence from the hierarchical organization of its neuronal hubs [163].

## 9. Mental Imagery and Sensorimotor Integration

Neuroimaging technologies (PET, fMRI), and also TMS, have demonstrated that mental imagery relies on much of the same neural machinery as perception, with the same modality [164]. It can also engage the mechanisms involved in memory, emotion, and motor control [165]. The early human visual cortex (areas 17, 18), for example, activates during imagery, indicating that visual imagery engages the same structures involved in perception within their respective modalities [164]. Similarly, imagery of emotional events activates the same brain region involved in specific perceptual experiences, including amygdala, insula and the autonomic nervous system [166]. Imagery is associated with all perceptual modalities. Odor imagery is associated with activity in the olfactory cortex, auditory imagery with bilateral activity in the secondary auditory cortices, tactile imagery with activity in the primary and secondary somatosensory areas, and motor imagery with activity in the premotor cortex and somatosensory cortex [167]. On the other hand, motor imagery depends on distinct neural mechanisms, although it occurs alongside visual imagery. Then, most of the neural processes that underlie modality perception are also used in imagery. Imagery, in many respects, can act as a substitute for perceptual stimulus “re-presenting” it.

Mental imagery can engage the motor system, which helps explain why ‘mental practice’ can enhance real performance. Imagining movements may not only stimulate relevant brain regions but also create connections among processes implemented in different areas, ultimately facilitating complex performances [168]. Imagery not only engages the motor system, but can also affect the body, similarly to actual perceptual experiences [164]. Ideomotor activity is fundamental to cognitive processes, including problem-solving, working memory, action simulation, and internal models. Through these mechanisms, anticipatory images are generated. Musicians, for example, often mentally rehearse their finger movements to optimize performance, also utilizing internal visual, auditory, and kinesthetic channels facilitating interpersonal coordination [168].

The ability to generate mental images of actions depending on internal and external factors make it possible to have accurate predictions about situations and other’s actions and mental states (social prediction) [169]. These mechanisms help the brain to create “internal notebooks” or working memories [170], and are the basis of cause–effect relationships that support the development of thought.

The involvement of neuronal networks with mirror properties in motor imagery is supported by numerous studies. The mechanism underlying this involvement stems from the properties of the mirror system. Whenever motor imagery is based on the observation of movement, the perception of sensory stimuli coupled with that movement induces the automatic activation of motor representations, since it overlaps with the execution of the same movement [49]. On the other hand, such a mechanism offers a good opportunity to understand the basis of movement suppression during motor imagery. Motor imagery involves both the planning and preparatory phases of movement, except for the execution phase, which is fully suppressed (i.e., without muscle activity) [171]. The ventral prefrontal cortex and anterior cingulate cortex have been found to participate in movement suppression during the preparation phase. The activation of the primary motor cortex is also more robust compared to the planning stage. Meanwhile, the premotor and supplementary motor areas are the primary regions activated, as they play crucial roles in the planning and preparation stages of motor control [171]. Visual or kinesthetic imagery engages the visual-related cortex and the inferior parietal lobe, respectively. The sensorimotor areas’ activation can be graded in relation to first- or third-person perspectives. Kinesthetic imagery, on the other hand, activates motor-related regions and the inferior parietal lobe [171]. When individuals visualize running, for instance, their bodies exhibit physiological responses similar to actual running, such as increased heart rate [172]. The supplementary motor area is activated during the mental rehearsal of processing of the spatial–temporal elements of movement sequences, highlighting the close relationship between mental representation and physical execution [112]. This activation occurs even in individuals with amputated limbs, where imagining movements with “phantom” limbs stimulates the motor cortex [173]. The motor imagery of a force production task has been reported to facilitate corticospinal excitability if immediately followed by physical practice, suggesting short-term functional changes in the primary motor cortex [174]. Mental imagery is proposed to be initiated from the frontal cortex, which then projects back to modality specific regions such as the visual cortex. People suffering with aphantasia, who are unable to voluntarily generate mental images, resultingly have deficits affecting all sensory modalities or that are modality-specific. On the contrary, subjects with extremely and atypically vivid mental imagery, hyperphantasia, show a visual specific modality overscore of self-reported imagery vividness. The strong functional connectivity in the resting state fMRI between the visual–occipital network and the prefrontal cortex of aphantasia-suffering subjects seems to suggest that imagery impairments can depend on a reduced connectivity between executive control regions and the visual cortex [167]. Then, despite being a cognitive process, motor imagery shares control mechanisms and neural substrates with actual movement and sensorimotor integration, offering insights into how movements are controlled by the brain. Defined as the mental rehearsal of movements without physical execution, motor imagery is used in sports, rehabilitation, and robotics.

The relationship between imagination and reality is manifested in the immediate neural adaptations that accompany physical exercise, but also in motivational and affective outcomes [121].

## 10. Implications

Motor imagery has gained attention as a tool for neurorehabilitation and brain–machine interfaces [BMIs]. Motor imagery, as the ability to identify the congruence between action observation and execution, forms the basis of action observation treatment, AOT, a therapeutic tool in neurodegenerative disease, in brain-lesioned patients, but also for the maintenance of motor skills in elderly people, or amelioration in sport performance [175]. To this respect, Bassolino et al. [176] have shown that action observation in maintaining cortical excitability is more efficacious than motor imagery in maintaining cortical excitability under immobilization conditions. Ideomotor training is a technique that involves mentally rehearsing a movement without physically executing it, with the goal of enhancing motor skills by engaging the brain’s motor planning and control systems [177]. The concept of mental practice implies the generation of a mental representation of movement while actively inhibiting motor execution. Neurophysiological and neuroimaging studies have demonstrated that motor imagery activates the same cortical motor areas involved in actual movement execution [178]. According to the “ideomotor principle”, the anticipation of sensory consequences can elicit a motor response [179,180]. This suggests that action sequences are planned based on their expected outcomes and that such “ideomotor planning” is modulated by domain-specific expertise. Indeed, proficient ideomotor planning has been proposed as a hallmark of skilled performance [181]. A more recent refinement of this concept is grounded in the theoretical framework of a common coding mechanism for action and perception [182], which aligns with the principles of sensorimotor integration discussed here. Ideomotor training specifically emphasizes mentally rehearsing a movement focusing on the brain’s ability to translate imaged movement into physical execution. It is often used in rehabilitation or skill acquisition. As mentioned, motor imagery involves mentally simulating or visualizing a movement without any actual physical movement with the goal to engage motor systems by imagining the movement in detail. It is frequently employed for improving performance in sports, rehabilitation, and brain training. The key difference between motor imagery and ideomotor training lies in their focus and application. While both involve mental rehearsal, ideomotor training is more explicitly about the mental-to-physical link and tends to be more directed toward preparing the body for movement through mental practice. Effective ideomotor training requires relaxation and focus, enabling individuals to “experience” themselves performing an action through internal visual, auditory, and kinesthetic sensations. This mental imagery stimulates the relevant muscles, boosting the impact of physical training. However, mental rehearsal alone cannot substitute for physical practice, as the kinesthetic aspects of movement are refined through actual execution. Ideomotor training can enhance muscle strength by up to 20%, underscoring the neural contribution to physical improvement [172]. The training technique of mental rehearsal, or visualization, is a vital aspect of athletic mental training. By vividly imagining performing a sport gesture, athletes can improve their skills without physical movement, provided that this mental practice is complemented by actual physical practice [172]. As a technique that involves visualizing movements without physical execution, ideomotor training, or mental practice, it enhances motor performance by activating motor areas and increasing muscle tone, supporting physical practice. Athletes and performers often use mental rehearsal to improve their skills, engaging all sensory channels to vividly imagine the action. Effective ideomotor training involves a combination of relaxation and concentration, allowing individuals to “perceive” themselves performing the action through internal visual, auditory, and kinesthetic channels. This mental imagery pre-activates the relevant muscles, enhancing the effectiveness of physical training. However, mental practice cannot replace the need for physical practice, as the kinesthetic aspects of movement are developed through actual execution. Mental rehearsal is particularly beneficial for complex gestures, where detailed visualization can enhance motor learning and execution. By imagining the movements vividly, individuals can improve muscle memory and coordination, complementing their physical training routines. The combination of mental and physical practice leads to better overall performance and skill acquisition [183]. The physiological mechanisms underlying ideomotor training involve the activation of motor cortical areas similar to those engaged during physical practice. Functional imaging studies have shown that mental rehearsal activates the primary motor cortex, premotor cortex, and supplementary motor area. This neural activation is thought to prime the motor system, making it more responsive to subsequent physical practice [184]. Ideomotor training can significantly improve motor learning and performance. Individuals who engage in mental rehearsal in addition to physical practice exhibit greater improvements in skill acquisition compared to those who rely solely on physical practice. This suggests that mental imagery can enhance the consolidation of motor skills in the brain. Combining mental and physical training is particularly effective for learning complex motor tasks. Mental rehearsal allows individuals to practice movements in their minds, refining their motor plans and improving coordination. When combined with physical practice, this approach leads to more efficient and effective motor learning, resulting in better performance and skill mastery. Also, it has been shown that the simulation of virtual reality may induce a modulation in the motor responses of physical reality [185]. Stroke patients treated for the simultaneous observation of repetitive movements displayed on a monitor peripheral nerve stimulation and the concurrent application of their own imagination of the sensations arising from physically performing the same action (AOT])have shown measurable improvements in dexterity [186]. Interestingly, studying the temporal accuracy of motor imagery of people with Parkinson’s disease [187] indicated that these patients imagine movements differently with respect to their more affected versus less affected side. The side-specific impairments result in an increased dependence on visual and cognitive processes to successfully execute motor imagery involving the more affected side, as a consequence of the lateralized deficits in the accurate processing of kinesthetic information [187]. Mental imagery is used by humans to intuitively solve most daily planning tasks. A novel algorithm based on “imaged” images has been proposed to enable robots to use existing scenarios to generate action plans. This method, called simulated mental imagery for planning (SiMIP), is different from deep learning and permits robot to operate through parsing process unlike predicting the entire scenes, thus enhancing the success rate of the plans whenever a feedback loop is incorporated [188]. Wearable robots, on the other hand, need to be embodied with wearers to facilitate motor and sensory reconstruction and enhancement. Control strategies targeting this embodiment are focused on integrating multisensory information (biomechatronic chips) to generate coherent control parameters configurations and improve human–robot interaction [189].

## 11. Conclusions

Investigating the functions of different cerebral regions and how they were intermingled, the history of our comprehension of sensory and motor systems has been changed to reach an integrated and merged ability of different functions raising from different areas of all cerebral tissue. Nowadays, there is growing evidence that the combined functions of all these regions have the possibility to harmonize sensorimotor integration until the emergence of new motor expressions, even of transferring this one to external avatars living in virtual reality [190]. Before we have completely described the sensorimotor basis of the “embodiment” of motor control, it is passed out and overtaken by imagery. Generating internal representations of actions is central to cortical motor function, and the external contingencies and motivational factors determine whether these action representations are transformed into actual (or virtual) actions through decisional processes that lead to action initiation. Related research reveals that motor imagery and action observation activate similar neural circuits that perform the actions themselves. This insight is pivotal for developing therapeutic strategies, especially for rehabilitation in patients with motor control disorders or brain injuries. By leveraging motor imagery, therapists can design interventions that stimulate relevant brain regions, potentially accelerating recovery and enhancing motor learning. Understanding these complex mechanisms can inform strategies for improving motor function and rehabilitation in individuals with motor impairments. The integration of sensory and motor functions is critical for effective interaction with the environment, enabling individuals to adapt their actions based on feedback. The role of predictive capabilities and procedural memory in motor learning highlights the importance of repetition and practice in acquiring and refining motor skills.

Rehabilitation programs can incorporate motor imagery and action observation to stimulate the brain’s natural learning processes without physical execution. This approach is particularly beneficial for stroke rehabilitation, where patients often struggle with regaining motor functions. Furthermore, these findings can inform the design of brain–computer interfaces (BCIs) that translate neural signals into actionable commands for prosthetic limbs, enhancing quality of life for individuals with severe motor impairments.

## Figures and Tables

**Figure 1 brainsci-15-00162-f001:**
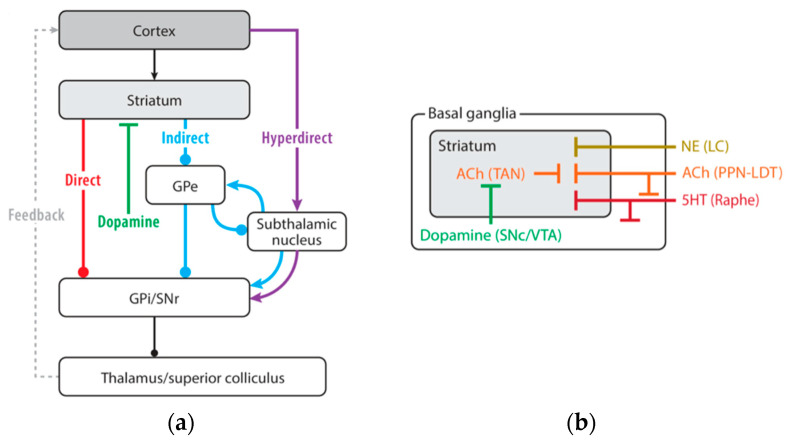
The basal ganglia pathway. (**a**) A simplified diagram of the cortico-basal ganglia–thalamus loop. (**b**) Multiple types of neuromodulatory inputs to the striatum. Abbreviations: 5HT, serotonin; ACh, acetylcholine; GPe, external segment of the globus pallidus; GPi, internal segment of the globus pallidus; LC, locus coeruleus; NE, norepinephrine; PPN-LDT, pedunculopontine nucleus–laterodorsal tegmental complex; SNc, substantia nigra pars compacta; SNr, substantia nigra pars reticulata; TAN, tonically active neurons; VTA, ventral tegmental area (modified from reference [85]. Copyright © 2023 Ding L. Licensed under CC BY 4.0).

**Figure 2 brainsci-15-00162-f002:**
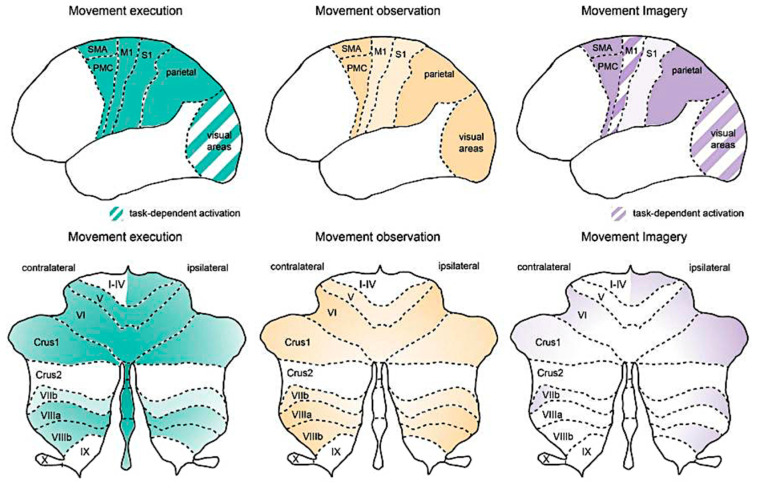
Activation in cortical and cerebellar regions during movement execution, observation, and imagery (from reference [107]: Copyright © 2023 Henschke and Pakan. Licensed under CC BY).

**Figure 3 brainsci-15-00162-f003:**
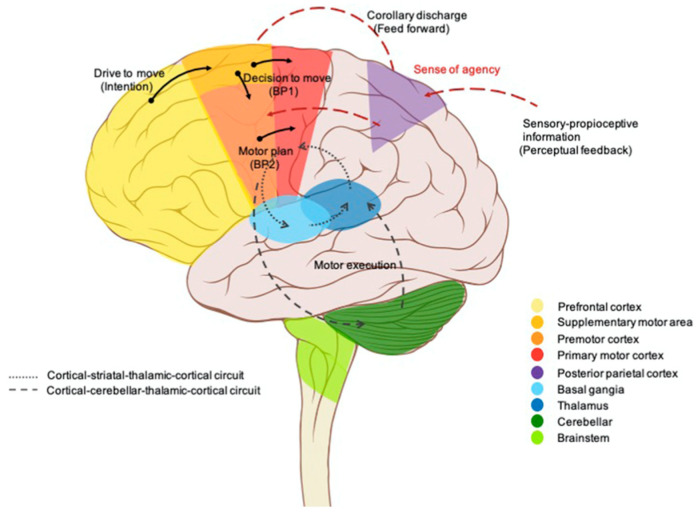
Schematic representation of voluntary movement circuitry. The intention to move is generated in the prefrontal cortex and limbic area. The complex movement sequences of movements are subsequently programmed in the presupplementary and supplementary motor areas. The premotor cortex is primarily involved in movements selection based on external information from the parietal cortex. The presupplementary and supplementary motor areas, along with the premotor cortex, produce the readiness potentials (BP1) as the expression of the brain preparing for the execution of the voluntary movement [110] that is transmitted through the motor cortex to the basal ganglia and cerebellum for motor control modulation. The processed information is then sent back to the motor cortex via the thalamus. Simultaneously, a corollary discharge (feedforward model) is generated and directed to the parietal cortex to compare with proprioceptive feedback, thereby creating a sense of agency. Ultimately, the neural signal exits the primary motor cortex (BP2) to the spinal cord and contralateral muscles, initiating the actual movement (modified from reference [109], Copyright © 2022 Virameteekul and Bhidayasiri. Licensed under CC BY).

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
