# Peer review of "Emergent Aspects of the Integration of Sensory and Motor Functions"

_brainsci, 2025, doi:10.3390/brainsci15020162_

Round 1

Reviewer 1 Report

Comments and Suggestions for Authors

Florio's paper is a narrative review that explores the neurophysiological mechanisms underlying sensorimotor integration. It offers a broad overview not only in terms of the mechanisms involved (including sensory information processing, motor learning, and movement generation) but also of brain structures, from cortical to subcortical ones. This interesting work could be relevant in increasing our understanding of these complex mechanisms. I have several suggestions for the Author: 

- Some chapters are extremely long, and it might be useful, in my opinion, to divide them into sub-sections to facilitate reading; 

- There are whole paragraphs without bibliographical references, and it would be useful to quote the sources to allow the reader to go deeper into the topics; 

- The Author rightly mentions Bayesian mechanisms for updating internal models of sensorimotor relationships. Given their relevance to the field under discussion, I suggest expanding the topic, i.e., explaining what neurons of internal representations and models, precision signals, and, above all, prediction errors are, explaining their usefulness in this field and also trying to locate them from a neurobiological point of view among the structures examined (e.g. doi: 10.1146/annurev-neuro-100223-121214);

- I suggest that a little exploited perspective, namely that of ‘embedded’ cognition, should be further explored. For example, we know that sensory inputs change motor responses, which in turn can critically influence the sensory information that will be acquired; these loops should be further explored, and their implications in terms of embedding should be explained; 

- Similarly, the role of attention should be explored in more depth; how can it modulate the sensory inputs gathered? This is particularly important if one considers the brain as a predictive machine and not as a passive collector of input from outside; 

- Lines 220-221: I suggest being more cautious in this formulation (it is likely that, like all CNS actions, it is a network that performs these actions);

- From the perspective of motor learning, I suggest expanding on other types of learning (e.g. use-dependent, reinforcement-based). This is particularly relevant when one considers that mono- and disynaptic connections have recently been shown between the cerebellum and basal ganglia and that the transfer of activities as learning increases has been shown in both regions (e.g. from cerebellar cortex to dentate nucleus to cortical areas, as for basal ganglia and cortical regions described). What does this convergence suggest? 

- A relevant citation to justify the difference in terms of ‘efficacy’ between execution, observation, and motor imagery could be this 10.1093/cursor/bht190. What are the implications for clinical practice?

- Another critical aspect that, in my opinion, should be mentioned, is the recent evidence linking the activity of the cerebellum to that of the mirror neuron network through cortical inhibitory interneurons, i.e. doi: 10.1016/j.neubiorev.2024.105830. This is particularly relevant for linking the described internal and learning models, as well as for providing a neurobiological circuit that can explain these functions. I therefore suggest mentioning this aspect and assessing its implications; 

- I was surprised, given its crucial role, by the absence of a reference to the mu rhythm (considered in the literature not only as the sensorimotor integration rhythm par excellence but also critically connected to the mirror network (again see doi: 10.1016/j.neubiorev.2024.105830). These aspects should be considered and, of note, it would be useful to assess their implications in pathological conditions, first and foremost stroke, which disconnects areas critically involved in these activities. For example, it has recently been shown with EEG studies that event-related desynchronization during action observation is an early predictor of recovery in subcortical stroke. Why? Could the function of the mirror system and, thus, sensorimotor integration be involved? 

- A final aspect that might be worth mentioning is the role of sleep in promoting sensorimotor integration. Again from an active inference theory (AIT) perspective, it has recently been proposed that sleep, particularly REM sleep, might be grounded in ‘defending’ cortical territories deprived of sensory input during sleep. How does this evidence fit into this context? Furthermore, dysfunction of these activities can, of note, lead to symptoms in various neurological and psychiatric contexts (e.g., positive symptoms, pain), for what reasons? 

Reviewer 2 Report

Comments and Suggestions for Authors

This review is generally well-organized and well-written, offering readers valuable insights into the mechanisms underlying sensory integration and motor control from a neuroscience perspective.

1. The major concern is that several key findings in this review lack appropriate references, such as Line 115-117, Line 183- 186, Line 233-235, Line 651-654 etc. Please carefully review the entire text and ensure that all findings or evidence are appropriately referenced with relevant literature. 

2. In terms of multisensory integration, the Bayesian theory provides a framework for understanding how the brain continuously updates the internal model to adapt to sensory inputs. As the author mentioned the Bayesian theory in the introduction part, the author should also incorporate a discussion of this theory with psychophysics or neurophysiological evidence. 

3. Line377-401: the author highlights the role of the premotor cortex during the initial stages of motor learning. However, the potential contributions of the prefrontal cortex in the early stages of motor learning are not discussed. A number of literature show the prefrontal cortex is activated in the early stage of motor learning. 

Reviewer 3 Report

Comments and Suggestions for Authors

The authors delved into the mechanisms underlying the sensory integration in the executive control of movement, encompassing ideomotor activity, predictive capabilities, and motor control systems. They reviewed the interplay between motor and sensory functions, highlighting the role of cortical and subcortical regions of the central nervous system in enhancing environmental interaction. The aspects of sensorimotor integration control and the implications are interesting and would contribute to clinical rehabilitation and assistive technology. There are a few minor issues as follows.

P3, L94-97: Please add references in each sentence.

P3, L134: Please add references in the sentence “Recent studies…”.

Figure 3: The text is blurred and difficult to read.

P14, L631: The abbreviation “PD” should not be used.

Round 2

Reviewer 1 Report

Comments and Suggestions for Authors

I thank the Author for the extensive work, which significantly improved the quality of this exciting paper. No further comments